# Discovery, Function, and Therapeutic Targeting of Siglec-8

**DOI:** 10.3390/cells10010019

**Published:** 2020-12-24

**Authors:** Bradford A. Youngblood, John Leung, Rustom Falahati, Jason Williams, Julia Schanin, Emily C. Brock, Bhupinder Singh, Alan T. Chang, Jeremy A. O’Sullivan, Robert P. Schleimer, Nenad Tomasevic, Christopher R. Bebbington, Bruce S. Bochner

**Affiliations:** 1Allakos Inc., Redwood City, CA 94065, USA; byoungblood@allakos.com (B.A.Y.); jleung@allakos.com (J.L.); rustom.falahati@gmail.com (R.F.); williamsmjas@gmail.com (J.W.); jschanin@allakos.com (J.S.); ebrock@allakos.com (E.C.B.); bsingh@allakos.com (B.S.); achang@allakos.com (A.T.C.); ntomasev@yahoo.com (N.T.); chrisbebbington@me.com (C.R.B.); 2Division of Allergy and Immunology, Department of Medicine, Northwestern University Feinberg School of Medicine, Chicago, IL 60611, USA; jeremy.osullivan@northwestern.edu (J.A.O.); rpschleimer@northwestern.edu (R.P.S.)

**Keywords:** Siglec-8, mast cells, eosinophils, monoclonal antibodies, glycan ligands, AK002, lirentelimab

## Abstract

Siglecs (sialic acid-binding immunoglobulin-like lectins) are single-pass cell surface receptors that have inhibitory activities on immune cells. Among these, Siglec-8 is a CD33-related family member selectively expressed on human mast cells and eosinophils, and at low levels on basophils. These cells can participate in inflammatory responses by releasing mediators that attract or activate other cells, contributing to the pathogenesis of allergic and non-allergic diseases. Since its discovery in 2000, initial in vitro studies have found that the engagement of Siglec-8 with a monoclonal antibody or with selective polyvalent sialoglycan ligands induced the cell death of eosinophils and inhibited mast cell degranulation. Anti-Siglec-8 antibody administration in vivo to humanized and transgenic mice selectively expressing Siglec-8 on mouse eosinophils and mast cells confirmed the in vitro findings, and identified additional anti-inflammatory effects. AK002 (lirentelimab) is a humanized non-fucosylated IgG1 antibody against Siglec-8 in clinical development for mast cell- and eosinophil-mediated diseases. AK002 administration has safely demonstrated the inhibition of mast cell activity and the depletion of eosinophils in several phase 1 and phase 2 trials. This article reviews the discovery and functions of Siglec-8, and strategies for its therapeutic targeting for the treatment of eosinophil- and mast cell-associated diseases.

## 1. Introduction

Siglecs (sialic acid-binding immunoglobulin-like lectins) are single-pass cell surface receptors that contain sialic acid-binding N-terminal V-set domains [1,2]. These receptors are predominately found on immune cells and most have immunoreceptor tyrosine-based motifs (ITIMs) that are involved in inhibitory cell signaling. The Siglec family of cell surface receptors have emerged as attractive therapeutic targets due to their restricted expression profile on immune cells and their immunomodulatory activities [3,4,5,6].

Mast cells and eosinophils are effector cells in the pathogenesis of many allergic and non-allergic diseases [7,8]. Of all the Siglecs, Siglec-8 is the only one found selectively on mast cells, eosinophils, and to a lesser extent, basophils [9,10]. Siglec-8 was discovered more than 20 years ago in a cDNA library generated from a subject with hypereosinophilic syndrome (HES) [9,10]. Our knowledge of the expression and function of Siglec-8 on mast cells and eosinophils has greatly expanded, as has our understanding of targeting Siglec-8 with monoclonal antibodies (mAbs) for potential therapeutic benefit. We review the discovery, function, and the potential for therapeutic targeting of Siglec-8 with mAbs.

## 2. Siglecs and the Discovery of Siglec-8

Siglecs, formerly called sialoadhesins or sialoadhesin factors (SAF) among other names, is a term that was adopted in 1998 to describe a subset of I-type lectins within the immunoglobulin gene superfamily that bind sialylated glycans and share certain structural motifs within their N-terminal and C-2 set domains [11]. At the time this nomenclature was adopted, only a few Siglecs were known. Since then, the list has expanded to include 10 human and 5 mouse Siglecs, belonging to Siglec-3 (CD33)-related subgroups, and 4 Siglecs were conserved between humans and mice (Siglec-1, Siglec-2 (CD22), Siglec-4, and Siglec-15). Besides the shared extracellular structural characteristics that define them as Siglecs, most possess tyrosine-based signaling motifs in the form of immunoreceptor tyrosine-based inhibitory or switch motifs (ITIMs and ITSMs, respectively). A small number of Siglecs lack such signaling domains because they have very short cytoplasmic tails. Examples include Siglec-14 and Siglec-16 in humans, Siglec-H in mice, and Siglec-15 in mice and humans. Each of these Siglecs lacking ITIM or ITSM domains contain a charged amino acid in their transmembrane domain that allows them to co-associate in the cell membrane with adapter molecules, such as DAP-12, to mediate SYK-dependent signaling.

Siglec-8 was discovered as a result of a joint effort between the Bochner and Schleimer laboratories at The Johns Hopkins University School of Medicine, along with scientists in the Department of Immunology, SmithKline Beecham Pharmaceuticals, and at Human Genome Sciences, Inc. Based on random, high-throughput EST (expressed sequence tag) sequencing, a clone from a cDNA library generated from a donor in the Bochner lab with a very high eosinophil blood count due to a form of HES was identified that possessed novel sequences with high homology to Siglec-7, Siglec-6, Siglec-5 and CD33. Simultaneously, the Crocker lab, via a separate collaboration with scientists at Human Genome Sciences, designed to identify novel Siglecs, discovered the same human eosinophil clone and sequence. Parallel investigations in both labs resulted in the publication of two papers in 2000 describing Siglec-8 (or sialoadhesin factor-2, SAF-2) as highly and selectively expressed on eosinophils [9], but also on human mast cells and very weakly on basophils [10]. As initially discovered, Siglec-8 contained a short cytoplasmic tail devoid of signaling motifs. This so-called short form was subsequently shown to result from a premature stop codon; the full-length mRNA encoded a form of Siglec-8 with a membrane-proximal ITIM and a membrane-distal ITSM that was subsequently shown to be the more predominant form of Siglec-8 [12,13].

Soon after Siglec-8 was discovered, mouse mAbs were generated [9,10] and the 2E2 Siglec-8 mAb clone was licensed in 2012 to Allakos, Inc. These have been the foundation of efforts at Allakos, Inc. to develop mAbs against Siglec-8 for clinical use.

## 3. Expression Pattern of Siglec-8

Siglec-8 is selectively expressed on human eosinophils, mast cells, and to a lesser extent, basophils, with ~18,000–22,000 Siglec-8 receptors/cell on eosinophils and mast cells and ~500 receptors/cell on basophils [14]. While it appears that the transcription factor Olig2 (Oligodendrocyte transcription factor 2) may participate in the control of Siglec-8 expression [15], more needs to be learned about how Siglec-8 expression is regulated. Unlike other receptors expressed on eosinophils, such as IL-5Rα, the expression of Siglec-8 is stable between blood and tissue compartments, suggesting Siglec-8 remains targetable by antibodies on both blood and tissue eosinophils [14]. Several groups have recently examined the expression pattern of Siglec-8 on eosinophils and mast cells during inflammatory disease states, including HES, asthma, eosinophilic esophagitis (EoE), eosinophilic gastritis (EG), and systemic mastocytosis (SM). The expression of Siglec-8 on blood eosinophils is similarly prominent in patients with eosinophilic diseases, including multiple variants of HES and EoE [16]. In addition, the levels of Siglec-8 remain stable on blood eosinophils after patients received treatment with prednisone or imatinib, despite significant decreases in absolute eosinophil counts [16]. Airway tissue eosinophils from bronchoalveolar lavage and sputum from subjects with asthma have similar levels of surface Siglec-8 as their blood eosinophils [17,18]. Likewise, Siglec-8 expression is comparable on gastrointestinal tissue eosinophils and mast cells from patients with EoE or EG, compared to tissues from control subjects without these diseases, despite elevated numbers and an activated phenotype of these cells in these diseases [19]. Similarly, mature bone marrow mast cells from patients with SM expressed Siglec-8 at levels similar to those observed in other tissues [20,21]. The expression of Siglec-8 remained low on basophils from patients with and without EoE or EG. Interestingly, a soluble form of Siglec-8 can be detected in the serum of some individuals using an ELISA, but the clinical significance and origin of this form of Siglec-8 remains unknown [16]. While flow cytometry is a powerful method for evaluating the expression of Siglec-8, as described above, we have found specificity differences between commercially available Siglec-8 antibodies due to the high sequence identity between some of the CD33-related family members. Using a Siglec-based cross-reactive ELISA, the Siglec-8 mAb clones FAB7975 (R&D Systems, Minneapolis, MN, USA) and 347104 (Biolegend, San Diego, CA, USA) were found to bind specifically to Siglec-8, whereas HPA012556 (Sigma and Atlas) was found to cross-react with multiple Siglecs, including Siglec-9, -7, and -12 (unpublished observations). These studies suggest that Siglec-8 expression is robust and stable on eosinophils and mast cells independent of disease state and tissue of origin.

Siglec-8 is not expressed by hematopoietic stem cells, or eosinophil or mast cell precursors. Based on in vitro studies in which cells were grown from precursors, Siglec-8 is only expressed during late stages of mast cell and eosinophil maturation [21,22]. Siglec-8 was not expressed by any eosinophil cell line tested and was expressed at low levels by the LAD2, LUVA, and HMC 1.2 mast cell lines [10,21,22] (unpublished observations). Like many CD33-related Siglecs, Siglec-8 is only expressed at the human and ape level, and is not detected on rhesus or cynomolgus monkey eosinophils [21]. Furthermore, Siglec-8 does not have a true mouse ortholog—the closest functional paralog in mice is Siglec-F, which is expressed on eosinophils (and other cells such as alveolar macrophages, tuft cells, granulocyte-macrophage progenitors and sometimes on certain tissue neutrophils that co-express Siglec-E) but not mast cells [23,24,25,26,27,28,29,30] (see Table 1 for Siglec-F and Siglec-8 comparison). Researchers in the Bochner laboratory and at Allakos, Inc. have developed strains of mice that selectively express Siglec-8 in the eosinophil compartment (SIGLEC8Eo), in the mast cell compartment (Mcpt5-Siglec8 and Cpa3-Siglec8), or on eosinophils, mast cells, and basophils (Siglec-8 transgenic) [19,31,32]. Strains that express Siglec-8 on specific immune cells rely on cell-specific or cell-selective Cre expression to remove a STOP cassette and allow for the discrimination between the effects of Siglec-8 on each cell population. Siglec-8 transgenic mice that express the human SIGLEC8 gene, including the putative promoter and regulatory elements, most accurately mimic the expression of Siglec-8 in humans. The SIGLEC8Eo strain has been crossed with the Siglec-F null strain, to create mice that express Siglec-8 but not Siglec-F [33]. These mice are useful for testing antibodies (see below) and glycomimetics that preferentially bind to Siglec-8, and for determining their specificity of targeting [34]. In addition, mice with humanized immune systems (engrafted with human thymus, liver, or hematopoietic stem cells) that express SCF, GM-CSF, and IL-3 (NSG-SGM3) generate human mast cells and eosinophils that express functional Siglec-8 [35].

## 4. Siglec-8 Function on Eosinophils and Mast Cells

Eosinophils and mast cells are innate immune cells that have broad roles in mediating and regulating acute and chronic tissue inflammation in allergic, proliferative, and inflammatory diseases. These cells are developmentally similar and frequently travel together to sites of inflammation through bi-directional crosstalk [8]. Historically, eosinophils and mast cells have been associated with allergic inflammation, during which their activation contributes to the development of type-2 inflammatory diseases, such as eosinophilic asthma, atopic dermatitis, and eosinophilic gastrointestinal diseases. In allergic inflammation, mast cells are primarily activated by crosslinking the FcεRI via IgE, which induces rapid degranulation and the release of preformed mediators, such as histamine, tumor necrosis factor (TNF), and proteases, as well as the subsequent release of de novo synthesized lipid mediators, cytokines, and chemokines [36]. In addition, eosinophils and mast cells have been implicated in non-allergic diseases, such as inflammatory bowel diseases, irritable bowel syndrome, functional dyspepsia, idiopathic pulmonary fibrosis, and chronic obstructive pulmonary disease. In these diseases, mast cells and eosinophils are likely activated by inflammatory mediators, such as cytokines, toll-like receptor ligands, and neuropeptides. Upon activation, mast cells and eosinophils release mediators that can attract or activate other immune cells and mediate acute and chronic inflammatory responses, such as vasodilation, plasma extravasation, smooth muscle contraction, stimulation of sensory nerves, tissue eosinophilia, epithelial barrier destruction, and fibrosis [37].

### 4.1. Anti-Eosinophil Activity

The expression pattern of Siglec-8 on peripheral blood and tissue eosinophils makes it an attractive therapeutic target for diseases associated with elevated eosinophils. Initial in vitro studies of Siglec-8 on eosinophils found that extensive antibody crosslinking induced cell death, which was caspase-dependent (see Figure 1 for summary of Siglec-8-mediated activity) [38]. Interestingly, the magnitude of Siglec-8 mAb-mediated eosinophil cell death increased when peripheral blood eosinophils were primed with IL-5, IL-33, or GM-CSF, and secondary crosslinking with anti-mouse antibody was no longer required [38,39,40]. Under these priming conditions, the mechanism of eosinophil cell death involved mitochondrial damage and reactive oxygen species (ROS) production, and was independent of caspase signaling [41,42,43]. Although the exact mechanism of Siglec-8-mediated cell death of eosinophils remains to be fully elucidated, the binding of a Siglec-8 mAb to cytokine-primed eosinophils activates a signaling pathway that involves β2 integrin-mediated adhesion; PI3K, Rac1, and MEK1/2 activity; and the generation of ROS via nicotinamide adenine dinucleotide phosphate oxidase, which ultimately results in cell death [43,44]. Consistent with Siglec-8 expression on tissue eosinophils, the exposure of human primary lung tissue, bronchoalveolar lavage, or sputum to AK002 ex vivo reduces the numbers of tissue eosinophils [14,18,39].

In addition to the antibody engagement of Siglec-8 on eosinophils, the effect of glycan ligands on Siglec-8-mediated cell death has also been evaluated. High-throughput glycan array screening led to the identification of two related potential Siglec-8 ligands: 6′-sulfo-sLe^x^ (NeuAcα2–3[6-*O*-sulfo]Galβ1–4[Fucα1–3]GlcNAc) and 6′-sulfated sialyl *N*-acetyl-d-lactosamine (NeuAcα2–3[6-*O*-sulfo]Galβ1–4GlcNAc); identical except for the presence or absence of the fucose [45,46]. These same two glycans are also selectively recognized by Siglec-F, although Siglec-F, unlike Siglec-8, recognizes additional multi-antennary α2,3-linked sialosides [25,45,47]. Further analysis demonstrated 6′-sulfo-sLe^x^ bound to human eosinophils as well as cells that express Siglec-8, and this interaction was inhibited by a polyclonal antibody against Siglec-8 [44,47]. The specificity of glycan binding was confirmed and defined using NMR spectroscopy [48]. The incubation of IL-5-primed blood eosinophils with either a Siglec-8 mAb or polymeric 6′-sulfo-sLe^x^ induced eosinophil cell death, although the glycan ligand was less effective [44]. Interestingly, the identification of the 6′-sulfo-sLe^x^ pharmacophores led to the development of ligand mimetics that bind Siglec-8 with higher affinity [34,49]. Although the search for endogenous glycan ligands is at an early stage, studies of human airway samples have identified keratan sulfate on aggrecan, and glycans on DMBT1 (deleted in malignant brain tumors 1) that bind Siglec-8 [50,51,52,53,54]. For example, a distinct isoform of DMBT1, carrying *O*-linked sialylated keratan sulfate chains (DMBT^S8^), was shown to be a high-avidity ligand for Siglec-8 that is found in the human airway mucus layer. However, the inhibitory activity of DMBT^S8^ was not evaluated on mast cells or eosinophils [53]. Studies are needed to determine the relevance of interactions between these glycan ligands and Siglec-8 in vivo, as well as any potential clinical relevance or association of these glycoforms (and the enzymes needed to synthesize them, such as specific sulfo- and sialyltransferases) with disease because the exact functions of natural Siglec-8 ligands are not known.

The activity of Siglec-8 mAbs in vivo was recently evaluated by several groups using transgenic and humanized mice that express functional Siglec-8. A single dose of a Siglec-8 mAb significantly reduced blood eosinophils in humanized mice with recombinant IL-5-induced eosinophilia [16]. To create a mouse model of eosinophilic gastroenteritis, Siglec-8 transgenic mice were given repeated intragastric injections of ovalbumin, which induced gastric and duodenal eosinophilia [19]. The administration of a Siglec-8 mAb with antibody-dependent cell-mediated cytotoxicity (ADCC) activity depleted blood eosinophils. In addition, Siglec-8 mAb treatment significantly reduced gastric, duodenal, and mesenteric lymph node eosinophils, as well as local and systemic inflammatory mediators, including CCL2, CCL5, and IL-9 [19].

Antibodies against Siglec-F also reduced the activity of eosinophils in mice, indicating that Siglec-F has similar functions to Siglec-8 on these cells. However, in mice that also express human Siglec-8 on their eosinophils, with or without endogenous Siglec-F, the administration of a mouse IgG1 anti-Siglec-8 mAb more effectively depleted eosinophils, albeit via ADCC, compared to several rat anti-Siglec-F antibodies. This is consistent with the findings that anti-Siglec-F antibodies have only modest effects on eosinophil death in vitro and depletion in vivo [33,55,56,57,58,59,60]. Therefore, anti-Siglec-8 mAb administration to the Siglec-8 knock-in mice may be a more effective approach to depleting eosinophils in mice [33]. Collectively, these data demonstrate that Siglec-8 mAbs have potent eosinophil-depleting activity in vitro and in vivo, and support the rationale for targeting Siglec-8 on eosinophils in the clinic.

### 4.2. Anti-Mast Cell Activity

Unlike the effect of the Siglec-8 mAb on eosinophils, crosslinking the receptor on unstimulated or activated mast cells with a mAb does not induce cell death. Instead, initial experiments demonstrated that incubation with Siglec-8 mAbs significantly inhibited the FcεRI-dependent release of histamine and prostaglandin D_2_ and the mast cell-dependent contraction of human bronchial smooth muscle rings in vitro [61]. Similarly, Siglec-8 mAbs inhibited the FcεRI-dependent degranulation of human primary lung tissue and CD34^+^ peripheral blood-derived mast cells without affecting mast cell numbers [18,62]. In addition to the inhibition of IgE-dependent activation, Siglec-8 mAbs significantly inhibited IL-33-mediated mast cell activation in vitro, as evidenced by the reduced IL-8 production and human neutrophil migration [62].

The ability to study Siglec-8 in vivo has greatly advanced our understanding of the effects of mast cell inhibition and the roles of mast cells in disease development. The administration of a Siglec-8 mAb to humanized mice populated with human mast cells protected them from systemic anaphylaxis-induced hypothermia [14]. In the Siglec-8 transgenic mouse model of eosinophilic gastroenteritis, the administration of the Siglec-8 mAb significantly reduced the infiltration of the stomach and small intestine by mast cells, and depleted eosinophils from blood and tissues [19]. In these mice, the reduction in mast cells following administration of the Siglec-8 mAb was delayed compared to the reduction in eosinophils in the same tissues. This finding indicates that the decrease in mast cell numbers was most likely due to the inhibition of mast cell recruitment or proliferation, rather than direct killing by antibody engagement in tissues. In support of this hypothesis, mice administered a Siglec-8 mAb had decreased levels of chemokines associated with the recruitment of mast cells [19], compared with mice given a control antibody.

Siglec-8 mAb administration was also shown to inhibit mast cells in mouse models of non-allergic diseases. In Siglec-8 transgenic mice undergoing a model of cigarette smoke-induced chronic obstructive pulmonary disease, Siglec-8 mAb administration attenuated airway inflammation and reversed the decline in lung function compared to mice dosed with an isotype control mAb [62]. These effects were associated with decreased numbers of lung mast cells, degranulating mast cells, and mast cell-derived proteases. In mouse models of acute and chronic bleomycin lung injury, the administration of a Siglec-8 mAb significantly reduced the airway infiltration of neutrophils, monocytes, and macrophages [62]. In addition, Siglec-8 mAb treatment reduced lung fibrosis, as determined by the Ashcroft score and collagen and TGF-β levels in bronchoalveolar lavage fluid [62]. The lung tissues from mice given the Siglec-8 mAb also had decreased levels of activated mast cells and mast cell-derived proteases.

The effects of Siglec-8 mAb on neutrophil infiltration have been investigated in mice with mast cell-mediated, IL-33-induced neutrophil influx. Mice given a Siglec-8 mAb and IL-33 had significantly reduced neutrophils and immune cell-recruiting cytokines, including IL-6, CCL2, CXCL2, IL-13, and TNF, compared with mice given a control antibody [62]. In addition to IL-33, Siglec-8 mAb treatment inhibited Substance P-induced mast cell activation, neutrophil infiltration, and cytokine production in Siglec-8 transgenic mice [63]. These data suggest that Siglec-8 mAb administration suppresses allergic and non-allergic inflammation mediated by mast cells or eosinophils, and that mast cells regulate the recruitment of immune cells, including neutrophils.

Although the exact intracellular signaling mechanisms involved in Siglec-8-mediated inhibition of mast cells have not been fully elucidated, studies with Siglec-8 ITIM-mutant transfectants revealed that the process requires the membrane-proximal ITIM tyrosine residue [61]. In vivo transcriptomic profiling of peritoneal mast cells activated with IL-33 and treated with a Siglec-8 mAb has yielded several insights [62]. Siglec-8 mAb treatment normalized the IL-33-activated mast cell transcriptome, in which the transcriptome of Siglec-8 mAb and IL-33-treated mast cells resembled that of the PBS-treated control mast cells. Pathway analysis revealed the downregulation of IL-33 signaling factors, including TNF, mTORC1 and IL-2–STAT5, in mice given the Siglec-8 mAb. Interestingly, the Siglec-8 mAb upregulated other receptors associated with the inhibition of mast cells, indicating that Siglec-8 regulates multiple signaling pathways. Additional studies are underway to delineate the exact signaling mechanisms of Siglec-8-mediated mast cell inhibition.

In addition to inducing ADCC or transducing an intracellular signal that leads to the death of eosinophils or the functional inhibition of mast cells, Siglec-8 can be exploited to deliver therapeutic cargo to these cell types via receptor-mediated endocytosis. In human eosinophils and mast cells in vitro, Siglec-8 is internalized in response to antibody ligation—a process that, in eosinophils, requires actin cytoskeletal rearrangement and the activities of tyrosine kinases and protein kinase C [64]. As a proof of concept, antibodies conjugated to the ribosome-inactivating toxin saporin induced the death of cells in which Siglec-8 ligation alone would not cause cell death, for example, in HMC-1.2 mast cell leukemia cells [64]. Whether this would be a safe and useful therapeutic approach worth pursuing remains uncertain.

## 5. Development of AK002, a Humanized Anti-Siglec-8 mAb

The selective inhibition of mast cell activity and the depletion of eosinophils by Siglec-8 antibody engagement provides a promising approach to the treatment of a wide range of allergic and inflammatory diseases associated with these cell types. AK002 is a humanized non-fucosylated IgG1 kappa antibody that was generated by the humanization of the mouse Siglec-8 mAb 2E2 clone (its precursor). AK002 is in clinical development for mast cell- and eosinophil-mediated diseases [14]. This antibody binds with high affinity to the extracellular domain of Siglec-8 and does not cross-react with other recombinant Siglecs. AK002 has the expected profile for a Siglec-8-specific antibody, binding selectively to mast cells, eosinophils and, at a lower level, to basophils in human blood and tissues, but not to the other cell types examined.

AK002 was generated as a non-fucosylated IgG1 antibody specifically to increase its binding affinity to FcγRIIIA, (CD16a), the Fc-receptor that mediates the ADCC activity of natural killer (NK) cells in peripheral blood [65]. Consequently, AK002 rapidly induces the NK cell-mediated ADCC of eosinophils in vitro and the selective depletion of eosinophils in peripheral blood leukocyte preparations, consistent with ADCC-mediated depletion. An IgG4 form of AK002, which lacks ADCC function, does not significantly deplete the eosinophils from peripheral blood leukocytes in the absence of exogenous cytokines, supporting the importance of FcγRIIIA-dependent ADCC for the activity of AK002 against peripheral blood eosinophils [16].

In addition to ADCC, AK002 induced the death of IL-5-activated eosinophils isolated from peripheral blood, highlighting an additional mechanism of eosinophil depletion within tissues in which CD16^+^ NK cells may not be prevalent [66], but where eosinophils may be primed by cytokines. When using dissociated human lung tissues from multiple donors, AK002 significantly reduced tissue eosinophil numbers. Mast cells in the dissociated lung tissue were not depleted [18], consistent with the view that the ADCC-inducing activity of AK002 may not be prominent in tissues. The lack of ADCC activity against tissue mast cells by AK002 is most likely due to the preponderance of the less cytotoxic, regulatory NK cell population (CD16^Lo^ CD56^Hi^) in most human tissues compared to the more cytotoxic, ADCC-inducing NK cell population (CD16^Hi^ CD56^Lo^) found in the blood [67]. Thus, the dual mechanism of action of AK002 against eosinophils, together with its inhibition of mast cell activation, provides a selective profile for the treatment of allergic and inflammatory diseases associated with these cell types.

## 6. AK002 Efficacy in Eosinophil- and Mast Cell-Mediated Diseases

AK002, now designated lirentelimab, is in clinical development for the treatment of multiple mast cell- and eosinophil-mediated diseases. Lirentelimab appears to have positive effects in patients with eosinophilic gastrointestinal diseases, chronic urticaria, severe allergic conjunctivitis, and indolent SM.

The first-in-human study of lirentelimab was ASIGMA, a phase 1, open-label trial in patients with indolent systemic mastocytosis (ISM). ISM is a rare disease characterized by clonal proliferation of, and excessive mediator release from, mast cells. Patients with ISM have an array of debilitating symptoms and substantial quality of life impairment. The study subjects received monthly infusions of lirentelimab over 6 months, with a starting dose of 1.0 mg/kg and subsequent doses of 1, 3, 6, or 10 mg/kg. Lirentelimab was well tolerated and produced consistent and substantial improvements in ISM symptoms and quality of life, measured by patient-reported outcome questionnaires [68].

Lirentelimab has also shown beneficial activity in patients with treatment-refractory chronic urticaria (CU). These patients have debilitating symptoms and many do not respond to other approved therapies, which include antihistamines and omalizumab. The CURSIG study was a phase 2a open-label trial that enrolled patients with chronic spontaneous urticaria (omalizumab naïve and omalizumab refractory) and chronic inducible urticarias (cholinergic and symptomatic dermographism). The patients received six monthly infusions of lirentelimab. Lirentelimab was well tolerated and showed substantial clinical activity for several patient-reported outcomes, in patients with different forms of treatment refractory CU and in patients refractory to extensive omalizumab treatment [69]. These results indicate that lirentelimab could be a first-line therapy for antihistamine-refractory CU.

Lirentelimab also substantially improves ocular signs and symptoms in patients with severe allergic conjunctivitis. Allergic conjunctivitis is an inflammatory disease characterized by extreme itching, pain, watering, redness, and swelling of the conjunctiva. In severe cases, corneal damage and permanent vision loss can occur. Eosinophil recruitment and mast cell activation contribute to symptoms, and allergic comorbidities are common. In a phase 1b open-label study, six monthly doses of lirentelimab were evaluated in patients with atopic keratoconjunctivitis, vernal keratoconjunctivitis, or perennial allergic conjunctivitis, and a history of topical or systemic corticosteroid use [70]. Lirentelimab was well tolerated and substantially improved ocular signs and symptoms of severe allergic conjunctivitis, with strong concordance between physician- and patient-reported outcomes. Lirentelimab produced substantial improvements in allergic comorbidities, reducing symptoms of atopic dermatitis, asthma, and rhinitis in 65%, 72%, and 69% of patients, respectively. Lirentelimab might therefore be a promising treatment for severe allergic conjunctivitis, as well as atopic dermatitis, asthma, and other atopic conditions.

Most recently, lirentelimab was evaluated in ENIGMA, a multi-center, randomized, double-blind, placebo-controlled phase 2 study in patients with active eosinophilic gastritis and/or eosinophilic duodenitis (EG/EoD) [71]. EG/EoD is characterized by gastrointestinal mucosal eosinophilia, chronic gastrointestinal symptoms, impaired quality of life, and a lack of adequate treatments. Mast cell activity has also been proposed to contribute to the pathogenesis of these conditions [19,72,73]. Patients with moderate to severe symptoms and biopsy-confirmed EG/EoD were randomly assigned to groups that received low-dose lirentelimab (*n* = 19; 1.0 mg/kg), high-dose lirentelimab (*n* = 20; 3.0 mg/kg), or placebo (*n* = 20). The combined lirentelimab groups had a mean 95% reduction in stomach and duodenal tissue eosinophils, compared with a 10% increase in the placebo group. There were significant reductions in EG/EoD total symptom scores in the lirentelimab group compared with placebo (reduction of 53% vs. a reduction of 24%, respectively; *p* = 0.0012). In addition, lirentelimab produced histologic and symptomatic improvements in patients with concomitant EoE. These data suggest that lirentelimab is a promising candidate for the targeted treatment of upper and lower EGIDs.

ENIGMA subjects were eligible to enroll in an open-label extension study, in which they received lirentelimab for as long as 20 months on a monthly basis. Fifty-eight of fifty-nine eligible subjects chose to enter the open-label extension study, and after 52 weeks of lirentelimab the mean symptom score was reduced by 68%, and 94% of subjects met the criteria for histologic remission. Long-term treatment with lirentelimab can therefore further reduce symptoms and sustain histologic remission.

Together, these data demonstrate the clinical utility of targeting Siglec-8 on mast cells and eosinophils with an mAb, and support the continued evaluation of lirentelimab in patients with mast cell- and eosinophil-mediated diseases. A phase 3 study of lirentelimab is underway in patients with eosinophilic gastritis and/or eosinophilic duodenitis (ENIGMA 2; NCT04322604), and a phase 2/3 study is underway in patients with EoE (KRYPTOS; NCT04322708).

## 7. Concluding Remarks

Since its discovery in a human cDNA library in the late 1990s and the first publications in 2000, our understanding of the function and expression of Siglec-8 on eosinophils and mast cells has steadily increased, along with the identification of a few endogenous glycoprotein ligands. Given the fact that Siglec-8 is not found in lower animal species, and that Siglec-F in mice is an inadequate surrogate for understanding Siglec-8 function, novel mouse models involving transgenic, knock-in or humanized mouse technologies have been developed and employed, and have reproduced most if not all of the in vitro biology, while delineating additional functions. Finally, since its founding in 2012, Allakos, Inc. has led the effort to humanize a mouse anti-human Siglec-8 antibody for clinical use. Multiple clinical trials using AK002 (lirentelimab), a non-fucosylated human IgG1 kappa antibody with enhanced ADCC activity, have shown efficacy across a spectrum of eosinophil- and mast cell-associated disorders, including several types of urticaria, ISM, severe allergic conjunctivitis, and EG/EoD. It is anticipated that lirentelimab will expand treatment options for multiple diseases involving mast cells and eosinophils, highlighting the broad function and clinical utility of targeting Siglec-8 with a mAb.

## Figures and Tables

**Figure 1 cells-10-00019-f001:**
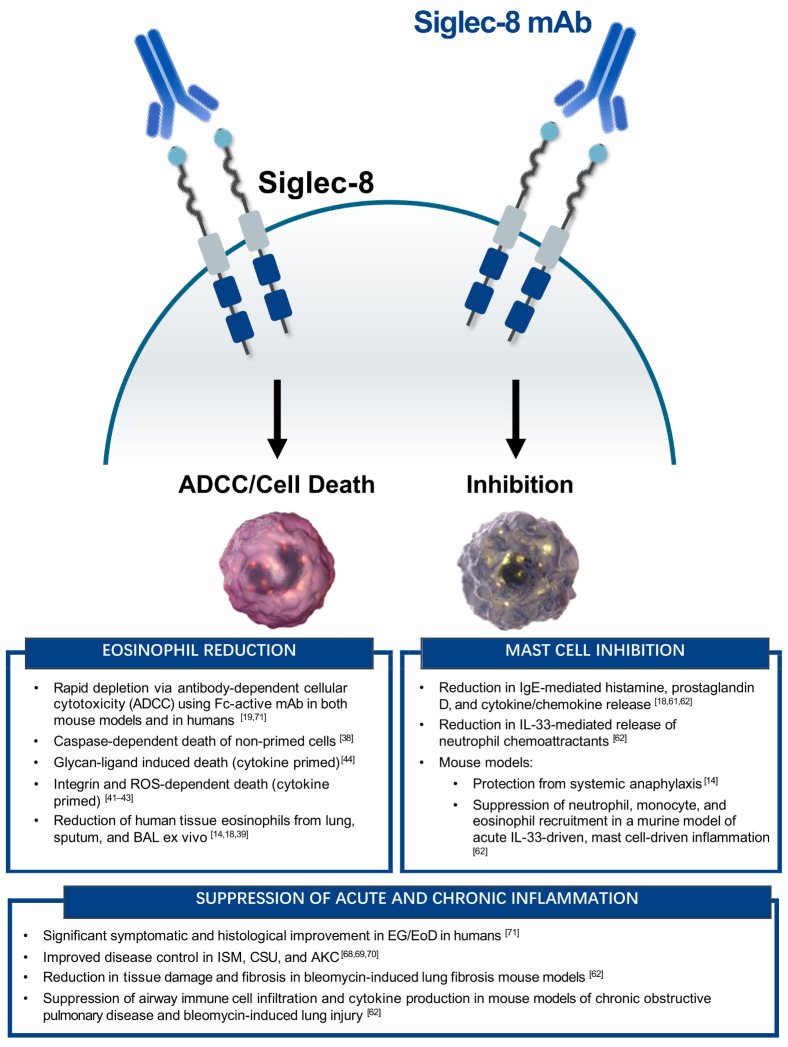
Summary of the in vitro and in vivo functions of Siglec-8 on eosinophils and mast cells. (**Left**) Eosinophil-specific activity of Siglec-8 mAbs or glycan-ligands. (**Right**) Mast cell-specific activity of Siglec-8 mAbs. (**Bottom**) Anti-inflammatory activity of targeting both eosinophils and mast cells with Siglec-8 mAbs. ADCC, antibody-dependent cellular cytotoxicity; AKC, atopic keratoconjunctivitis; CSU, chronic spontaneous urticaria; EG/EoD, eosinophilic gastritis/eosinophilic duodenitis; ISM, indolent systemic mastocytosis.

**Table 1 cells-10-00019-t001:** Comparison of Siglec-8 and Siglec-F.

	Siglec-8	Siglec-F
**Surface Expression**		
Eosinophils	Yes	Yes
Mast cells	Yes	No
Basophils	Yes; weak	No
Alveolar macrophages	No	Yes
Neutrophils	No	Sometimes
T cells	No	No or minimal
Monocytes	No	No
Intestinal tuft/M cells	No	Yes
Expression is at least in part regulated by the transcription factor Olig2	Yes	Unknown
**Ligands**		
6’-*S*-Sialyl-LacNac	Yes	Yes
6’-*S*-Sialyl-Lewis X	Yes	Yes
Tri and tetra-antennary bisected glycans containing α2,3-linked terminal sialic acid	No	Yes
Sialylated keratan sulfate chains on human aggrecan	Yes	Unknown
Sialylated keratan sulfate chains on human DMBT1	Yes	Unknown
Mouse Muc5b glycans	No	Yes
9-*N*-(2-naphthyl-sulfonyl)-Neu5Acα2-3-[6-*O*-sulfo]-Galβ1-4GlcNAc (6’-*O*-sulfo (NSA)Neu5Ac)	Yes	Yes
6′-sulfo-sialyl Lewis X mimetic retaining the neuraminic acid core, but with a carbocyclic mimetic of the Gal moiety and a sulfonamide substituent in the 9-position	Yes	Unknown
**Function**		
*Eosinophils* in vitro: *Non-cytokine primed*		
Crosslinking with antibody induces eosinophil death in non-cytokine-primed cells	Yes; modest	Yes; weak
Death that is caspase-dependent	Yes	Yes
Death that is integrin- and ROS-dependent	No	No
Death that is NADPH oxidase-dependent	No	No
Death is associated with mitochondrial membrane damage	Yes	Yes
Receptor internalized after ligation	Yes	Yes
*Eosinophils* in vitro: *Cytokine primed*		
Crosslinking with antibody or multivalent ligand induces eosinophil death in cytokine-primed cells	Yes; marked	Yes; weak
Death that is caspase-dependent	No	Yes
Death that is beta-2 integrin- and ROS-dependent	Yes	No
Death that is NADPH oxidase-dependent	Yes	No
Death that is associated with mitochondrial membrane-damage	Yes	Yes
Role for SHP-1 phosphatase in cell death	No	No
Role for MAP kinases in cell death	Yes	Unknown
*Mast cells* in vitro		
Crosslinking induces cell death	No	Not applicable
Inhibition of IgE receptor-mediated degranulation	Yes	Not applicable
Inhibition of IL-33-stimulated responses	Yes	Not applicable
Receptor internalized after ligation	Yes	Not applicable
Internalization of a toxic payload after ligation causes cell death	Yes	Not applicable

Siglec, sialic acid-binding immunoglobulin-like lectins; DMBT1, deleted in malignant brain tumors 1; Gal, galactose; ROS, reactive oxygen species; MAP, mitogen-activated protein.

## Data Availability

The unpublished data mentioned in this review are available on request from the first author due to privacy. The data are not yet publicly available because they have not yet been published. Otherwise, no new data were created or analyzed in this review.

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
