# Peer review of "Discovery, Function, and Therapeutic Targeting of Siglec-8"

_cells, 2020, doi:10.3390/cells10010019_

Round 1

Reviewer 1 Report

This is a very well written paper and explains current concepts regarding Siglec-8 and its immunobiology and disease-significance with great clarity.  I learned a lot while reviewing this manuscript and agree with many of the conclusions reached by the authors regarding the promise of Siglec-8 as a potential target for allergic and inflammatory disorders which have posed many challenges to management, in spite of many advances. The application of this anti-Siglec-8 monoclonal antibody AK-002 to such seemingly diverse disorders as urticaria, eosinophilic gastroenteropathy and severe vision-threatening ocular disorders is particularly exciting.  I do have a few minor critiques which are summarized below:

1.  The lack of effect of glucocorticoids and TK-inhibitors on Siglec-8 expression is interesting in the face of decreased eosinophil numbers and altered mast cell function is a curious phenomenon and may need some explanation

2.  Why is it that there are differences in expression of this molecule in basophils and mast cells--is it that the molecule is shed or lost during differentiation of these cell types?

3.  The reason for the apparent lack of differences in expression between tissue and blood stream cellular expression of Siglec-8 is also unclear and what role this may play in disease pathogenesis

4.  The cellular distribution is different between mice and men, and one therefore wonders if mice studies can be extrapolated directly to man. Siglec-F/8 expression on neutrophils of mice but not in humans is likewise of interest.  Is this related to alterations in stem cell biology and gene expression between species?

5. Figure 1 needs a better explanation in the legend--as the role/effects of the MoAb are not explained properly.  The figure combines mice and human data making the interpretation difficult, but some of this is explained later under the Lirentelimab data (CIU, AKC, ISM etc) but references not cited or explained properly. This could be modified slightly perhaps to make these aspects more clear

6.  Lines 158-160 are not very clear--as to how these studies were accomplished

7.  Are there any interactions between Siglec-8 and type 2 cytokine or mediator/granule expression by mast cell/eosinophil lineage cells?.  In other words, do Siglec-8 molecule signaling motifs and downstream pathways modulate mast cell-eosinophil interactions (cognate or paracrine)?

8.  A slightly better focus on Siglec-8 ligands (expression, cell types)  may be of benefit to the neophyte or casual reader--especially aggrecans, glycans on DMBT1 etc--how these are expressed in tissue or correlate to disease pathogenesis, eosinophil death and inhibition of mast cell activation in inflammatory states is not immediately clear 

9.  The effects of Siglec-8 antibody anaphylaxis and mast cell mediator inhibition, profound histological effects on EoE, lung fibrosis (TGF-beta axis), neutrophil biology, acute phase cytokines ands chemokines are exciting but suggest pleomorphic effects bordering on immunosuppression--although these were mixed mice and human studies.  This aspect raises concerns regarding safety

10.  On a philosophical level--With rising health care costs, and the high cost burden of immunobiologicals, where would the authors think Siglec-8 inhibition would be most worthwhile?.  Many of the current biologicals for atopic and inflammatory disorders are effective, but clinicians are uncertain when to stop them and hence many patients remain on these drugs seemingly for a long time.

Author Response

This is a very well written paper and explains current concepts regarding Siglec-8 and its immunobiology and disease-significance with great clarity. I learned a lot while reviewing this manuscript and agree with many of the conclusions reached by the authors regarding the promise of Siglec-8 as a potential target for allergic and inflammatory disorders which have posed many challenges to management, in spite of many advances. The application of this anti-Siglec-8 monoclonal antibody AK-002 to such seemingly diverse disorders as urticaria, eosinophilic gastroenteropathy and severe vision-threatening ocular disorders is particularly exciting.

            Reply: Thanks for these comments, and for those that follow. We appreciate the overall inquisitiveness of this reviewer, but as you will read, there are still a lot of unanswered questions. In revising the paper, we have tried to highlight these issues.

I do have a few minor critiques which are summarized below:

  1. The lack of effect of glucocorticoids and TK-inhibitors on Siglec-8 expression is interesting in the face of decreased eosinophil numbers and altered mast cell function is a curious phenomenon and may need some explanation.

            Reply: We agree that more needs to be learned about the regulation of Siglec-8 expression. To date, there is literally just one published paper that explores intracellular pathways regulating Siglec-8 expression, implicating Olig2 as a contributing transcription factor, but this alone does not explain the entire story because a) its absence only partially affected Siglec-8 expression on eosinophils, b) it did not explain Siglec-8 expression on mast cells, and c) basophils were not studied (Hwang, S.M.; Uhm, T.G.; Lee, S.K.; Kong, S.K.; Jung, K.H.; Binas, B.; Chai, Y.G.; Park, S.W.; Chung, I.Y. Olig2 is expressed late in human eosinophil development and controls Siglec-8 expression. J Leukoc Biol 2016, 100, 711-723). Besides this, the only thing that is known to impact Siglec-8 surface expression is engagement of the receptor, e.g., by antibody, a process that results in its internalization (O'Sullivan, J.A.; Carroll, D.J.; Cao, Y.; Salicru, A.N.; Bochner, B.S. Leveraging Siglec-8 endocytic mechanisms to kill human eosinophils and malignant mast cells. J Allergy Clin Immunol 2018, 141, 1774-1785). While we can detect soluble Siglec-8 in serum in some individuals using an ELISA, the clinical significance and origin of this form of Siglec-8 remains unknown (Legrand, F.; Cao, Y.; Wechsler, J.B.; Zhu, X.; Zimmermann, N.; Rampertaap, S.; Monsale, J.; Romito, K.; Youngblood, B.A.; Brock, E.C., et al. Sialic acid-binding immunoglobulin-like lectin (Siglec) 8 in patients with eosinophilic disorders: Receptor expression and targeting using chimeric antibodies. J Allergy Clin Immunol 2019, 143, 2227-2237).  See new text on lines 77-79 and lines 94-96, as well as one edit near the top of Table 1.

  1. Why is it that there are differences in expression of this molecule in basophils and mast cells--is it that the molecule is shed or lost during differentiation of these cell types?

            Reply: As stated above, more needs to be learned on how Siglec-8 expression is regulated. One possible explanation is the presence of upstream regulatory elements surrounding the Siglec-8 promoter that could drive cell specific expression. In support of this, the Siglec-8 transgenic mice developed at Allakos were generated by insertion of the Siglec-8 gene including putative upstream regulatory elements and these mice recapitulate not only the selective expression but also the relative levels of expression of Siglec-8 on eosinophils, mast cells, and basophils (Youngblood, B.A.; Brock, E.C.; Leung, J.; Falahati, R.; Bochner, B.S.; Rasmussen, H.S.; Peterson, K.; Bebbington, C.; Tomasevic, N. Siglec-8 antibody reduces eosinophils and mast cells in a transgenic mouse model of eosinophilic gastroenteritis. JCI Insight 2019, 4, doi:10.1172/jci.insight.126219).  See reply to question #1 as well as new text on lines 77-79.

  1. The reason for the apparent lack of differences in expression between tissue and blood stream cellular expression of Siglec-8 is also unclear and what role this may play in disease pathogenesis.

            Reply: Sorry for causing any confusion. Our point, which is hopefully now been made clearer in the revised text, was not about disease pathogenesis per se but more about how tissue cells, by maintaining Siglec-8 expression, remain targetable by antibody. Other eosinophil surface molecules, like the IL-5 receptor, are lower in cells derived from other tissue compartments like the airway, giving rise to speculation that such cells may become less susceptible to drugs that target IL-5 or its receptor.  See new text on lines 80-81.

  1. The cellular distribution is different between mice and men, and one therefore wonders if mice studies can be extrapolated directly to man. Siglec-F/8 expression on neutrophils of mice but not in humans is likewise of interest. Is this related to alterations in stem cell biology and gene expression between species?

            Reply: Confidentially, you have touched on a bit of a nerve here. Before Allakos took up the banner of moving Siglec-8 antibodies into the clinic, numerous scientists at multiple pharmaceutical companies, as well as granting agencies, poo-pooed the idea based on their interpretation of Siglec-F mouse data. But to answer your question, there is one paper (Bolden, J.E.; Lucas, E.C.; Zhou, G.; O'Sullivan, J.A.; de Graaf, C.A.; McKenzie, M.D.; Di Rago, L.; Baldwin, T.M.; Shortt, J.; Alexander, W.S., et al. Identification of a Siglec-F+ granulocyte-macrophage progenitor. J. Leukoc. Biol. 2018, 104, 123-133) showing that there is a Siglec-F+/CD11b+/IL-5R- granulocyte-macrophage progenitor in normal mouse bone marrow, and this population exists even in eosinophil-deficient mice. Furthermore, these cells have granulocyte‐macrophage progenitor-like developmental potential both in vitro and in vivo, and are transcriptionally distinct from the classically described granulocyte‐macrophage progenitor population. An additional publication suggests that tumor-infiltrating neutrophils in mice have a unique phenotype including dual expression of both Siglec-F and Siglec-E (Pfirschke, C.; Engblom, C.; Gungabeesoon, J.; Lin, Y.; Rickelt, S.; Zilionis, R.; Messemaker, M.; Siwicki, M.; Gerhard, G.M.; Kohl, A., et al. Tumor-promoting Ly-6g(+) SiglecF(high) cells are mature and long-lived neutrophils. Cell Rep 2020, 32, 108164). So far, these findings are unique to mice. Similarly, why mouse mast cells fail to express Siglec-F, or why mouse alveolar macrophages express Siglec-F, while human macrophages do not express Siglec-8, remains unknown. But overall, we certainly agree that Siglec-F ≠ Siglec-8, which is why we used this Review as an opportunity to highlight both similarities and differences in Table 1.  See edits on lines 111-112 including the addition of two new references.

  1. Figure 1 needs a better explanation in the legend--as the role/effects of the MoAb are not explained properly.  The figure combines mice and human data making the interpretation difficult, but some of this is explained later under the Lirentelimab data (CIU, AKC, ISM etc) but references not cited or explained properly. This could be modified slightly perhaps to make these aspects more clear

            Reply: We apologize for the confusion. We have revised the figure by adding more text to the legend and including references.  See edits to the figure, which now includes references, plus edits to the legend on lines 149-154.

  1. Lines 158-160 are not very clear--as to how these studies were accomplished.

            Reply: We have added some text for clarification on line 173.

  1. Are there any interactions between Siglec-8 and type 2 cytokine or mediator/granule expression by mast cell/eosinophil lineage cells? In other words, do Siglec-8 molecule signaling motifs and downstream pathways modulate mast cell-eosinophil interactions (cognate or paracrine)?

            Reply: The short answer is that direct interactions between eosinophils and mast cells and their modulation by Siglec-8 has not been studied directly in vitro. The longer answer is that while IL-5 priming facilitates Siglec-8 engagement-induced cell death in vitro, it does not stimulate eosinophil degranulation via Siglec-8 engagement (Nutku-Bilir, E.; Hudson, S.A.; Bochner, B.S. Interleukin-5 priming of human eosinophils alters Siglec-8 mediated apoptosis pathways. Am. J. Respir. Cell Mol. Biol. 2008, 38, 121-124 and Carroll, D.J.; O'Sullivan, J.A.; Nix, D.B.; Cao, Y.; Tiemeyer, M.; Bochner, B.S. Sialic acid-binding immunoglobulin-like lectin 8 (siglec-8) is an activating receptor mediating beta2-integrin-dependent function in human eosinophils. J. Allergy Clin. Immunol. 2018, 141, 2196-2207). For mast cells, regardless of cytokine priming or not, Siglec-8 engagement does not kill them in vitro (Yokoi, H.; Choi, O.H.; Hubbard, W.; Lee, H.S.; Canning, B.J.; Lee, H.H.; Ryu, S.D.; von Gunten, S.; Bickel, C.A.; Hudson, S.A., et al. Inhibition of FcεRI-dependent mediator release and calcium flux from human mast cells by sialic acid-binding immunoglobulin-like lectin 8 engagement. J. Allergy Clin. Immunol. 2008, 121, 499-505). However, the clinical trials with AK002 show ≥95% depletion of eosinophils from the blood, and a similar degree of depletion in GI tissues, accompanied by ≈20-25% reduction in mast cells in these same GI tissues (Dellon, E.S.; Peterson, K.A.; Murray, J.A.; Falk, G.W.; Gonsalves, N.; Chehade, M.; Genta, R.M.; Leung, J.; Khoury, P.; Klion, A.D., et al. Anti-Siglec-8 antibody for eosinophilic gastritis and duodenitis. N Engl J Med 2020, 383, 1624-1634). Whether the depletion of eosinophils in vivo by AK002 directly or indirectly reduces mast cells in these tissues is an interesting but unanswered question. Because so little is known, and what is mentioned above is already discussed in the paper, we have not made any new edits here. However, see edits related to Question #1 from Reviewer #2 below.

  1. A slightly better focus on Siglec-8 ligands (expression, cell types) may be of benefit to the neophyte or casual reader--especially aggrecans, glycans on DMBT1 etc--how these are expressed in tissue or correlate to disease pathogenesis, eosinophil death and inhibition of mast cell activation in inflammatory states is not immediately clear.

            Reply: The reviewer has touched on another area of importance. To date, all of the published work on endogenous ligands for Siglec-8 have focused on the human airway, since that was what we were funded to explore. Phase 1 of these studies was to identify ligands, and so far, using either surgical specimens and postmortem samples, certain sugars (glycans) on aggrecan isolated from tracheal connective tissue, and probably identical glycans on DMBT1 generated in submucosal glands of the nasal mucosa, have been identified. Whether there are disease-related variants in the airway, and whether there are additional ligands in other tissues, is currently being investigated. In parallel, levels of the specific sulfo- and sialyltransferases necessary to generate these ligands in the Golgi during posttranslational modification of these scaffold proteins are also being examined to see if they are altered in disease.  See new text on lines 193-195.

  1. The effects of Siglec-8 antibody anaphylaxis and mast cell mediator inhibition, profound histological effects on EoE, lung fibrosis (TGF-beta axis), neutrophil biology, acute phase cytokines and chemokines are exciting but suggest pleomorphic effects bordering on immunosuppression--although these were mixed mice and human studies. This aspect raises concerns regarding safety

            Reply: We also have been intrigued by the broad anti-inflammatory activity demonstrated with anti-Siglec-8 antibodies. Interestingly, pre-clinical studies have shown that Siglec-8-mediated inhibition only occurs when mast cells are activated, and the extent of inhibition never exceeds that of homeostatic basal activity. In addition, over 428 patients have been dosed with lirentelimab to date with no sign of immunosuppression even with long-term dosing seen in the ENGIMA extension study where patients received antibody for over 1 year (Dellon, E.S.; Peterson, K.A.; Murray, J.A.; Falk, G.W.; Gonsalves, N.; Chehade, M.; Genta, R.M.; Leung, J.; Khoury, P.; Klion, A.D., et al. Anti-Siglec-8 Antibody for Eosinophilic Gastritis and Duodenitis. N Engl J Med 2020, 383, 1624-1634, doi:10.1056/NEJMoa2012047). While we appreciate the reviewer’s comments, we have not made any new edits here.

  1. On a philosophical level--With rising health care costs, and the high cost burden of immunobiologicals, where would the authors think Siglec-8 inhibition would be most worthwhile? Many of the current biologicals for atopic and inflammatory disorders are effective, but clinicians are uncertain when to stop them and hence many patients remain on these drugs seemingly for a long time.

            Reply: Targeting Siglec-8 with lirentelimab is currently being studied in moderate to severe eosinophilic gastrointestinal diseases (EGIDs). These conditions are typically characterized by a long, tortuous journey for the patients who have to endure chronic GI symptoms for years. Currently there are no FDA approved treatments for EGIDs. While it is not possible at this time to comment on costs or durability/duration of treatment at this stage of development, the authors believe that safe and effective targeted therapies for chronic conditions with limited therapeutic options will contribute meaningfully towards improving patients' quality of life. While we appreciate the reviewer’s comments, we have not made any new edits here.

Reviewer 2 Report

The manuscript by Youngblood et al on Siglec-8 is an interesting review to read, from its discovery to recent clinical trials with the anti-Siglec-8 antibody, AK002. I only have a few minor comments.

  1. Since the effect of anti-Siglec-8 on eosinophils and mast cells are quite different, i.e., induction of ADCC in eosinophils but not mast cells, it would be of interest to have a discussion about this. Although the mechanisms might not be clarified, the authors may have som thoughts about this.
  2. Figure 1 legend: The abbreviations should be in alphabetical order.
  3. Line 288: Mast cells are not necessary "activated", in the meaning that a ligand has bound to a receptor an activated the cells. I think activated is misleading. It might be an intrinsic, not necessarily extrinsic, effect. I would recommend to omit "activated".   

Author Response

The manuscript by Youngblood et al on Siglec-8 is an interesting review to read, from its discovery to recent clinical trials with the anti-Siglec-8 antibody, AK002. I only have a few minor comments.

  1. Since the effect of anti-Siglec-8 on eosinophils and mast cells are quite different, i.e., induction of ADCC in eosinophils but not mast cells, it would be of interest to have a discussion about this. Although the mechanisms might not be clarified, the authors may have some thoughts about this.

            Reply: Thanks for these comments and suggestions. In humans, ADCC is thought to be mainly driven by NK cells. Several groups have shown that peripheral blood NK cells (CD16Hi CD56Lo) have more cytotoxic activity compared to tissue NK cells (CD16Lo CD56Hi) which have more of a regulatory, less cytotoxic phenotype (Poznanski SM and Ashkar AA (2019) What Defines NK Cell Functional Fate: Phenotype or Metabolism? Front. Immunol. 10:1414. doi: 10.3389/fimmu.2019.01414). Consistent with this, lirentelimab induces ADCC of both eosinophils and mast cells in vitro in the presence of peripheral blood NK cells (unpublished observations). However, since mast cells are only found in tissue and Siglec-8 is not expressed on immature mast cell progenitors, lirentelimab does not reduce mast cells in vivo nearly as much as it does eosinophils, most likely due to low levels of cytotoxic NK cells in the tissue.

See additional new text on lines 295-298 along with one new reference.

  1. Figure 1 legend: The abbreviations should be in alphabetical order.

            Reply: This has been corrected.

  1. Line 288: Mast cells are not necessary "activated", in the meaning that a ligand has bound to a receptor an activated the cells. I think activated is misleading. It might be an intrinsic, not necessarily extrinsic, effect. I would recommend to omit "activated".

            Reply: We agree and have removed the word “activated” on line 307.